# Effects of brief mindfulness training on smoking cue-reactivity in tobacco use disorder: Study protocol for a randomized controlled trial

**Linlin Cheng**[1,2☯], **Miaoling Luo**[1,2☯], **Jie Ge**[2,3], **Yu Fu**[1,2], **Quan Gan**[1,2,4‡], **Zhuangfei Chen**[1,2‡]*

**1** Medical School, Kunming University of Science and Technology, Kunming, China, **2** Brain Science and Visual Cognition Research Center, Medical School of Kunming University of Science and Technology, Kunming, China, **3** Students Counseling and Mental Health Center, Kunming University of Science and Technology, Kunming, China, **4** Faculté de médecine, Université Paris-Saclay, Le Kremlin-Bicêtre, France

☯ These authors contributed equally to this work.
‡ QG and ZC also contributed equally to this work.
\* chen.zhf@outlook.com

**Data Availability Statement:** No datasets were generated or analysed during the current study. All relevant data from this study will be made available upon study completion.

## Abstract

### Background

The prevalence of Tobacco Use Disorder (TUD) represents a significant and pressing global public health concern, with far-reaching and deleterious consequences for individuals, communities, and healthcare systems. The craving caused by smoking cue is an important trigger for relapse, fundamentally hindering the cessation of cigarette smoking. Mindfulness interventions focusing on cue-reactivity was effective for the treatment of related dependence. Brief mindfulness training (BMT) meets the short-term needs for intervention but the effects still need to be examined. The objective of the present study is to investigate the impact of BMT intervention on smoking cue-reactivity among Chinese college students with TUD, to uncover the dynamic models of brain function involved in this process.

### Method

A randomized control trial (RCT) based on electroencephalography (EEG) was designed. We aim to recruit 90 participants and randomly assign to the BMT and control group (CON) with 1:1 ratio. A brief mindfulness training will be administered to experimental group. After the intervention, data collection will be conducted in the follow-up stage with 5 timepoints of assessments. EEG data will be recorded during the smoking cue-reactivity task and 'STOP' brief mindfulness task. The primary outcomes include subjective reports of smoking craving, changes in EEG indicators, and mindfulness measures. The secondary outcomes will be daily smoking behaviours, affect and impulsivity, as well as indicators reflecting correlation between mindfulness and smoking cue-reactivity. To evaluate the impact of mindfulness training, a series of linear mixed-effects models will be employed. Specifically, within-group effects will be examined by analysing the longitudinal data. Additionally, the effect size for all

**Funding:** This study was supported by the National Natural Science Foundation of China (NSFC) (Nos. 32060196, 82360271 and 82201597) and Yunnan Ten Thousand Talents Plan Young and Elite Talents Project (YNWR-QNBJ-2018-027, YNWR-QNBJ-2018-056), and Innovation team of Stress and disorder in nervous system in Yunnan (202305AS350011).

**Competing interests:** This study was supported by the National Natural Science Foundation of China (NSFC), Yunnan Ten Thousand Talents Plan Young and Elite Talents Project and Innovation team of Stress and disorder in nervous system in Yunnan. The funders had no role in study design, data collection and analysis, decision to publish, or preparation of the manuscript. The financial support was provided in forms of research materials. The authors have declared that no competing interests exist.

statistical measurements will be reported, offering a comprehensive view of the observed effects.

## Discussion

The current study aims to assess the impact of brief mindfulness-based intervention on smoking cue-reactivity in TUD. It also expected to enhance our understanding of the underlying processes involved in brain function and explore potential EEG biomarkers at multiple time points.

## Trial registration

Trial registration number: ChiCTR2300069363, registered on 14 March 2023. Protocol Version 1.0., 10 April 2023.

## Introduction

### Tobacco epidemic

As a major determinant of preventable health consequences, Tobacco Use Disorder (TUD) is a problematic pattern of tobacco use that leads to significant health and economic burdens [1]. There are 1 billion people confronting the problem worldwide. Cigarette smoking can increase the incidence and mortality rates of various diseases [2–4], causing an annual death of over 8 million people worldwide [6]. The health costs and other economic losses caused by smoking had already reached $1.4 trillion annually a decade ago [7]. Since the WHO Framework Convention on Tobacco Control came into force, more and more countries have adopted effective measures to control smoking. From 2007 to 2021, the global average smoking prevalence has reduced from 22.8% to 17.0% [8].

In China, the population of individuals with TUD exceeds 300 million [9]. The smoking rate among people over 15 years reach to 26.6%, for male is up to 50.5% [8]. Due to the delay effect, it is projected that the disease burden for both individuals and the overall population of Chinese adult males with TUD will increase in the future. Additionally, those who begin smoking at a younger age are at a higher risk of morbidity and mortality from major diseases [10]. Without effective intervention, the number of tobacco related deaths is expected to rise to 2 million per year by 2030 and 3 million per year by 2050 [9].

In July 2019, the National Health Commission of China released the "Healthy China Action (2019–2030)" [11], which aims to decrease the smoking rate among individuals aged 15 and above to 20% by 2030.

Besides increasing individual awareness of the detrimental effects of smoking and taking steps to quit, it is also essential to establish smoke-free environments and implement practical measures at personal, family, societal, and governmental levels. This tobacco control initiative also emphasizes the promotion of effective short-term smoking cessation interventions, aiming to provide support to more individuals who are in the process of quitting smoking. Moreover, post-adolescence is a crucial transitional phase from substance misuse and experimentation to addiction, primarily due to the ongoing development of the brain during this period. Hence, this stage of 'pre-addiction' is deemed a critical juncture for identifying brain circuits that could contribute to future challenges in life [12]. The necessity of

strengthening smoking control during the young adult stage has been put on the agenda, and there is an urgent need to explore more targeted smoking cessation methods in this population.

## Smoking craving

Craving is defined as a strong desire to pursue an addictive substance [12]. Smoking craving is the main symptom of tobacco dependence [14] and has been treated as an important indicator for smoking cessation treatment [15], and reflected by objective measures [16].

The acquisition and maintenance of nicotine addiction is a complex procedure. Individuals with TUD may experience an activation of their craving and consumption impulses in response to specific environmental contexts, situational cues, and stimuli [17–18]. When the brain fails to regulate reward impulses efficiently, cravings can disrupt one's self-control and their ability to consider future consequences, thereby increasing their propensity to make impulsive decisions [20, 21]. It is essential to note that the concept of cue-reactivity extends beyond the traditional stimulus-response framework. In the case of smoking, cue-reactivity encompasses not only the direct association between the cue and the smoking behavior but also the complex cognitive and emotional processes underlying craving and relapse. Simultaneously, the correlation among addictive cue stimuli is dynamic and can be influenced by interactions between multiple stimulus. Consequently, increased exposure and interventions are necessary during the process of extracting and reconsolidating addictive memories effectively [22]. Investigating cue-reactivity within the framework of TUD necessitates a comprehensive grasp of various elements. These include attentional bias, intensity of cravings, and subjective responses to cues, as well as the interaction of internal and external condition [22, 24]. Moreover, due to the ubiquitous nature of addictive stimuli, simply suppressing cravings can lead to a greater rebound effect [25], which is a significant factor contributing to relapse and treatment resistance [26]. Therefore, some scholars argue that therapies solely targeting the addiction cycle are often challenging to succeed, highlighting the necessity for innovative treatment approaches.

## Mindfulness intervention for TUD

Mindfulness can be defined as a conscious and open process of focusing on the present moment with curiosity [26]. The non-reactive, present-moment awareness associated with mindfulness can increase the likelihood of perceiving each moment as a unique and independent event, unaffected by past experiences. This can help individuals accept psychological and physiological discomfort, which conceptualized as a process of "interoceptive desensitization" [28] rather than relying on substances for relief. Furthermore, mindfulness can serve as a type of refined exposure since many clinical issues arise from inflexible efforts to avoid unpleasant internal experiences [28]. Compared to previous studies, incorporating mindfulness exposure may better capture the diverse influences of cue, emotion, and classical conditions on smoking behavior. By focusing on the downstream relationships between each factor and smoking behavior, it allows for a more comprehensive understanding of the variations observed [30]. As a result, mindfulness-based interventions have been explored within the context of treatment for TUD, with promising future in helping to prevent addiction and relapse [31]. It can directly affect the neurocircuitry of cravings, reducing addictive behaviour by decreasing subjective cravings, attention, and related physiological indicators of cue-reactivity to addictive substances [32]. At the same time, compared to active control with relaxation training [32], mindfulness can also help individuals manage negative emotions and physical sensations, allowing them to dissociate their emotional responses from their cravings for smoking [30].

However, due to the long-term intervention period of mindfulness, the demand for short-term practice is increasing. A few studies have confirmed that a single intervention of mindfulness and a 5-minute intervention can improve health outcomes [34, 35]. Brief mindfulness intervention can reduce smokers' response to craving [36], improve emotional health [30], increase 7-day point-prevalence abstinence [37], and reduce smoking frequency and the number of cigarettes [38]. Moreover, the participation in mindfulness exercises was related to the abstinence rate in the follow-up study. At the same time, it has been confirmed that BMT can reduce subjective reported negative emotions, smoking cravings, and cigarette use in natural situations [38], indicating the effectiveness of this method in smoking cessation treatment. Meanwhile, in terms of brain mechanisms, both traditional 8-week mindfulness [40] and short-term mindfulness can affect interoceptive processing, to be more targeted in regulating brain function in cue-reactivity [32], making it an ideal intervention strategy for cue-reactivity. Accordingly, exteroceptive processes also serve as targets for interventions of mindfulness training [41] for it acting a critical role in conditioned cue response in addiction. The balance between interoceptive and exteroceptive is also emphasized [41]. Among the non-clinical population, both short-term and high-dose mindfulness training programs demonstrate similar effects in alleviating psychological distress [43].

In line with prior research [43], our previous trial has also observed that standard 8-week mindfulness can lead to functional changes in brain areas involved in interoceptive processing such as the insula [45] as well as in the executive control system, including the cingulate gyrus, parietal lobe, and prefrontal lobe [46, 47]. However, most EEG studies examining mindfulness have consistently found a positive correlation between improved emotional states and increased alpha power [48]. Additionally, internalized attention has been linked to heightened theta activity in the frontal midline [49].

In recent years, there has been increasing evidence suggesting that low-theta EEG coherence in the brains of smokers may serve as a potential biomarker for smoking cue reactivity and be able to predict addictive behaviors [50]. Mindfulness intervention was associated with lower P3 amplitude and reduced response inhibition [50]. Neurofeedback training also revealed deactivated EEG activity patterns (P300 amplitudes) associated with smoking cue reactivity [20]. The prefrontal-insula circuit could be a noninvasive imaging biomarker for monitoring treatment efficacy [51]. Microstate class C (duration) may indicate cue-induced cigarette craving, while class D (duration and contribution) could reflect the relationship between cue-elicited activation of the dorsal attention network and years of smoking [16].

Therefore, we hypothesize that brief mindfulness intervention can reduce smoking cue-reactivity in the population of TUD and can be reflected in subjective craving reports and changes in EEG functional indicators. To capture the diverse aspects of smoking craving, we will assess the EEG power spectrum, ERP components, microstate, and functional connectivity indicators before and after intervention. These will be analysed in combination with subjective craving and smoking behavior. By using source analysis and rely on the time resolution advantages of EEG, we can further explore the regulation effects of smoking cue-reactivity and related brain functional mechanisms of TUD following a brief mindfulness intervention.

This study aims to investigate the impact of a 7-day brief mindfulness practice under the treatment paradigm for substance dependence on smoking cue-reactivity through a RCT design, incorporating a follow-up of multi-session intensive training and online homework tracking of self-guided practices. Previous studies have shown that mindfulness, when compared with control group, does not reduce craving. However, it does influence individuals' readiness and willingness to respond to cues, encouraging the use of mindfulness strategies to manage smoking urges and derive benefits from cue exposure paradigms [53].

On one hand, it may make up for the shortcoming of long mindfulness and promote acceptance in the population with TUD. On the other hand, the exploration of brain function mechanisms can also provide more targeted intervention strategies for smoking cessation, which will be of great value for promoting the clinical application of brief mindfulness in tobacco addiction intervention. Therefore, the study has the potential to offer quantitative evidence supporting the decoupling of cue-reactivity and cravings in young adults with TUD.

## Materials and methods

### Study design and participants

This study will adopt a randomized parallel control trial design, and about to recruit participants by combining online and offline advocation. Following previous mindfulness research methodologies, we will integrate relaxation training as an active control in this study to enhance its superiority design.

Participants who meet the experimental requirements will join the study by following a sequence that includes baseline data collection, intervention training, post-intervention data collection, and follow-up time points. After the completion of baseline data collection in two time points (-T1, T0; exp., EEG data in -T1), participants will be randomly assigned to the BMT and CON group respectively, to complete the corresponding training for a week. Except for the baseline, a total of 5 subsequent data collections are anticipated after intervention (T1) along with follow-up at four timepoints (T2-T5). During the follow-up period, participants will receive monthly intensive training sessions, as well as online homework tracking for self-guided practices every two weeks. The study sites will be the current living environment in the university for the participants.

### The inclusions and exclusions

Inclusion criteria: (1) Right-handed. (2) Smoking 10 or more cigarettes per day. (3) Smoking history of at least 1 year. (4) Exhaled carbon monoxide (CO) 10-7ppm or lower before assessment (abstinent from smoking cigarette for 3 h prior to every visit). (5) Age between 18 and 40 years old. (6) Normal or corrected to normal vision. (7) Normal mental and physical health conditions (PHQ-9 [54–55] total score <20, GAD-7 [57, 58] total score <11). (8) Meets the diagnostic criteria for tobacco use disorders in DSM-5 assessed by clinical structured interview [58].

Exclusion criteria: (1) Asthma, contact dermatitis, or allergies to silicone (exclude those allergic to materials undergoing EEG). (2) Recent use of corticosteroid medications within the past 3 months. (3) Practiced any meditation, yoga, tai chi, or qigong for more than 20 hours in the past year or lifetime, participated in meditation or yoga retreats, and attended any meditation courses. (4) Other situations not suitable for EEG (e.g., metal implants, severe head trauma and electrode allergy). (5) PHQ-9 total score ≥20. (6) GAD-7 total score ≥11. (7) Adhere to specific religious beliefs that and therefore unable to participate in meditation as required by the course. (8) Currently participating in similar trials or other neurophysiological studies. (9) On medication assisted treatment for tobacco use disorder before or during the study.

Handedness and vision criteria ensure task completion and minimize variability. Potential participants will also be screened by evaluating whether they met the aforementioned inclusion criteria and exclusion criteria after completing the demographic questionnaire and screening scales, concluding PHQ-9 and GAD-7, followed by written informed consent, as stipulated by the Declaration of Helsinki (2008).

## Sample size and power calculation

In the current study, the primary indicator used to assess smoking urges will be self-reported subjective measures. Based on previous research conducted on similar populations [60], we have defined a reduction score of 4 on the Brief Questionnaire of Smoking Urges (QSU-B) as the expected decrease in smoking craving. Therefore, in the experimental group after the intervention, we anticipate observing a $\Delta_{QSU-B}$ of 4. We intend to conduct multiple measurements on the participants before and after training, as well as during the online assessment period for the smoking cue-reactivity task, and anticipate achieving 80% power (1-beta = 0.8), at the significant level of 5% (alpha = 0.05, two sided). The Cohen's d was estimated with by the formula of $d = \Delta_{QSU-B}/SD_{estimate}$, where SD was 7.65 according to reference study, resulting in an estimated sample value of 34. For the estimation of the sample size of the experimental group, it is required to have no less than 34 people. Considering a natural dropout rate of 30%, the sample size for experimental group should be no less than 45 people. To ensure proper comparability, an additional 45 participants who match in demographic data and smoking severity will be assigned to the control group. Consequently, a total of 90 participants are required for the present study.

## Recruitment

We will recruit participants by combining online and offline methods, including lectures, posters, and distributing flyers in the classroom, also through Qzone, WeChat Moments, campus BBS and school's bulletin board. After filling in the registration form (collecting demographic characteristics information, i.e., age, gender, contact information, smoking years, average number of cigarettes smoked per day, etc.), the recruited participants will be informed of the general process of the study, and the participants need to sign the written informed consent according to their wish. After confirming their participation in the study, participants will be scheduled to fill out relevant trait scales, including the Smoking Addiction Scale (FTND), the Five Facet Mindfulness Questionnaire (FFMQ), the Smoking Desire Scale Simplified (TCQ-SF), the Edinburgh Handedness Inventory (EHI), and the Positive And Negative Affect Scale (PANAS) and so on.

## Baseline assessment

The data collection in baseline will start from status-related scales. After filling out the scale, EEG data for pre-intervention will be recorded from all participants. Participants will be asked not to smoke for at least 3 hours before collecting EEG data. Upon arrival in the laboratory, participants will be required to report when the last cigarette smoked and exhale CO concentration need to be measured by using a Bedfont smoke detector. Only those with a CO value between 10-7ppm or lower before EEG data collection can be proceed to the EEG process, otherwise, they need to reschedule another time to come to the laboratory for EEG data collection. Data collection for all participants will take approximately two weeks.

The details are as follows: (1) Firstly, a time slot appointment form for collecting EEG will be sent to the group chats for participants. They will need to arrange a specific time for data collection according to their own schedule; (2) Notify participants who have already made appointments in advance of not smoking before collecting EEG data; (3) After the participants arrived at the laboratory, they need to record when their last cigarette smoked before starting EEG process, and accept CO testing; If the CO detection does not meet the standards, the current EEG collection cannot be performed and a new collection time needs to be scheduled. Only those who meet the EEG collection standards are required to collect baseline EEG data after filling out the scale.

## Randomization and group allocation

Participants will be allocated into BMT and CON group. Each participant who meets the inclusion criteria will be assigned a unique number for this study. The statistician will then randomly assign the participants to the two groups in a 1:1 ratio, matched in demographic data and smoking severity (i.e., duration and daily consumption). Then retain the grouping information until the baseline data collection is completed, and the grouping results will be fed back to the researchers by the statistician before the intervention training begins. The random sequence generation website (https://www.random.org/) will be used for the process.

## Data collection

FTND, TCQ-SF, FFMQ, EHI, PHQ-9, GAD-7 and other scales as well as participant registration forms (for the collection of demographics characteristics information) will be produced, and released via WenJuanXing, a Chinese professional platform for online survey (https://www.wjx.cn/). Online data collection will also be implemented on the platform to facilitate the unified management of the data. After collection is finished, the data will be downloaded for the preparation of analysis. EEG data acquisition will be conducted by using a 64-channel EEG acquisition system subsequently. To those participants failed to finish data collection, we will send a reminder through WhatsApp in time.

## Procedures

The overall process is shown in Figs 1 and 2 and S1 File. After completing the recruitment, scales will be distributed to all participants and combined with an exhale CO detection. For eligible participants, they will be required to sign the written informed consent according to

| | STUDY PERIOD | | | | | | |
| --- | --- | --- | --- | --- | --- | --- | --- |
| | Enrolment | Allocation | Post-allocation | | | | Close-out |
| TIMEPOINT* | $-t_1$ | 0 | $t_1$ | $t_2$ | $t_3$ | $t_4$ | $t_5$ |
| **ENROLMENT:** | | | | | | | |
| Eligibility screen | X | | | | | | |
| Informed consent | X | | | | | | |
| Baseline assessment | X | X | | | | | |
| Allocation | | X | | | | | |
| **INTERVENTIONS:** | | | | | | | |
| [Brief mindfulness training] | | ●——— | | | | | |
| [Relaxation training] | | ●——— | | | | | |
| **ASSESSMENTS:** | | | | | | | |
| [Demographic data] | X | | | | | | |
| [Primary outcome variables] | X | X | X | X | X | X | X |
| [Secondary outcome variables] | X | X | X | X | X | X | X |
| [Daily records during intervention] | | ●——— | | | | | |

**Fig 1. SPIRIT schedule of enrolment, interventions and assessment.** Note: -t1, 0, baseline; t1, post intervention; t2-t5, 1-month, 3-month, 6-month, 1-year follow-up.

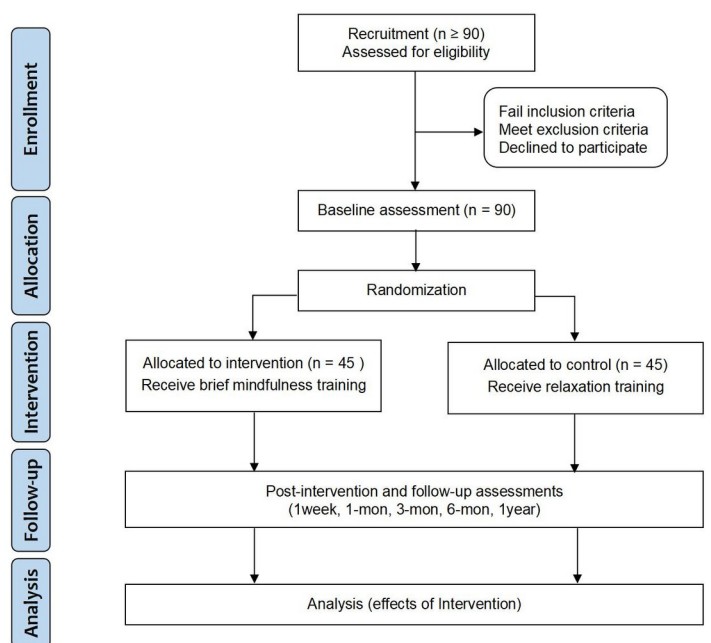

**Fig 2. SPIRIT diagram.**

participants' will, and then randomly allocated into BMT and CON groups. The baseline data from all participants will be collected at two different time points within a week. A 7-day brief mindfulness intervention will be conducted immediately after the baseline assessment. Post-intervention (i.e., 1-week timepoint) data will be collected when the training is completed, and follow-up data in the other 4 timepoints will be collected sequentially.

## Brief mindfulness intervention

A 7-day mindfulness training will be organized after baseline data collection, and guided by professional teachers from the school's Students Counseling and Mental Health Center. The themes of brief mindfulness training in each day are listed in Table 1. The audio guidance for exercise will be distributed by researchers after the end of each day's training, lasting for one week. The BMT group need to carry out 'STOP' mindfulness exercises, while CON group carry out relaxation exercises in spare times as homework in each day for intervention.

**Table 1. Overview of the brief mindfulness training.**

|       | Mindfulness practices |
| --- | --- |
| **Day 1** | Fundamentals of mindfulness and breathing Meditation, 'STOP' practice of mindfulness |
| **Day 2** | Mindfulness meditation, body scan, and five senses awareness |
| **Day 3** | Mindfulness meditation, sharing of life pleasant events, and mindfulness sanding yoga |
| **Day 4** | Mindfulness meditation, forgiveness and self-acceptance, and self-compassion practice |
| **Day 5** | Gratitude and attention to others, mindfulness walking practice |
| **Day 6** | Outdoor meditation |
| **Day 7** | The integration of mindfulness in daily life |

*Audio guidance for 'STOP' practice was delivered for daily exercise in spare time.

Participants can choose different time periods for multiple exercises throughout the day, and fill out both the daily smoking records and the daily practice records. They need to record the extra practice duration (in minutes) and evaluate their seriousness and compliance level (1–10 points) independently. The daily smoking behaviour (number of cigarettes smoked, total in 7 days) will also be collected by delivered forms.

To address the concern that brief mindfulness training might not produce observable long-term effects due to limited exposure time, we have referred to the intervention settings in existing literature research [41, 61]. After 7 days of intervention, a monthly intensive training session will be organized, lasting for one day (2 hours) each time. Following the training session, homework practicing time will be collected every two weeks, with follow-up evaluations of the intervention effect scheduled at the 1-month, 3-month, and 6-month timepoints. Group training is structured as collective activities, which enhance the effectiveness of the intervention. The brief mindfulness intervention with multiple sessions satisfies the current demand for shorter intervention duration among college students, while also providing the potential for accumulating sufficient exposure time.

### Control group

Control group will be assigned to relaxation training, conducted at the same time each day as the experimental group during the 7-day period of intervention. The training will include watching documentaries of natural scenery without any other intervention, for which has been suggested a generic relaxation effect of video interventions on autonomic regulation [62, 63]. The audio guidance and learning materials for both groups will be uploaded to the group chats separately, participants can practice at any time after the intervention period. The total number of meditation/relaxation sessions completed as homework for both groups will be recorded. The intensive training schedule during all follow-up periods is kept the same for both groups.

### Post-intervention data collection

After the intervention training (T1), we will send a time slot appointment form for EEG collection to the participants' group chats, and the participants need to arrange their own time for the appointment, following the same requirement of smoke cessation and EEG data collection procedure as in baseline. Follow-up data collection will be conducted for participants at consequent four time points, i.e., T2-T5 (1-month, 3-month, 6-month, and 1-year). EEG data and scale data will all be collected with the same setting in T1-T5 (Fig 3).

### 'STOP' mindfulness practice

'STOP' stands for four mindful actions: 'Stop', 'Take a breath', 'Observe', 'Proceed'. It is a helpful aid in becoming more mindful of our body, behaviour and emotion on a daily basis. The following is the adapted instruction of practicing STOP in confronting smoking cessation based on that developed by Liao [63]:

S = **stop;** Remind yourself to STOP. Whatever you are doing right the moment (e.g., looking for cigarettes or a lighter, ready to remove the cover of pack, hold the cigarette, putting cigarette into mouth etc.), pause for a minute.

T = **take;** Take a deep breath. The body will become relax and quiet in its natural breathing rhythm. Breath is the anchor of the mind and body, also present moment and awareness.

O = **observe;** Observe what is happening for you in this moment: except for breath, what is your thoughts, feelings and emotions (e.g., feel distracted, anxious or nervous). What do you notice in your body (e.g., feel tired or unease on any part of your body)? You can be aware of

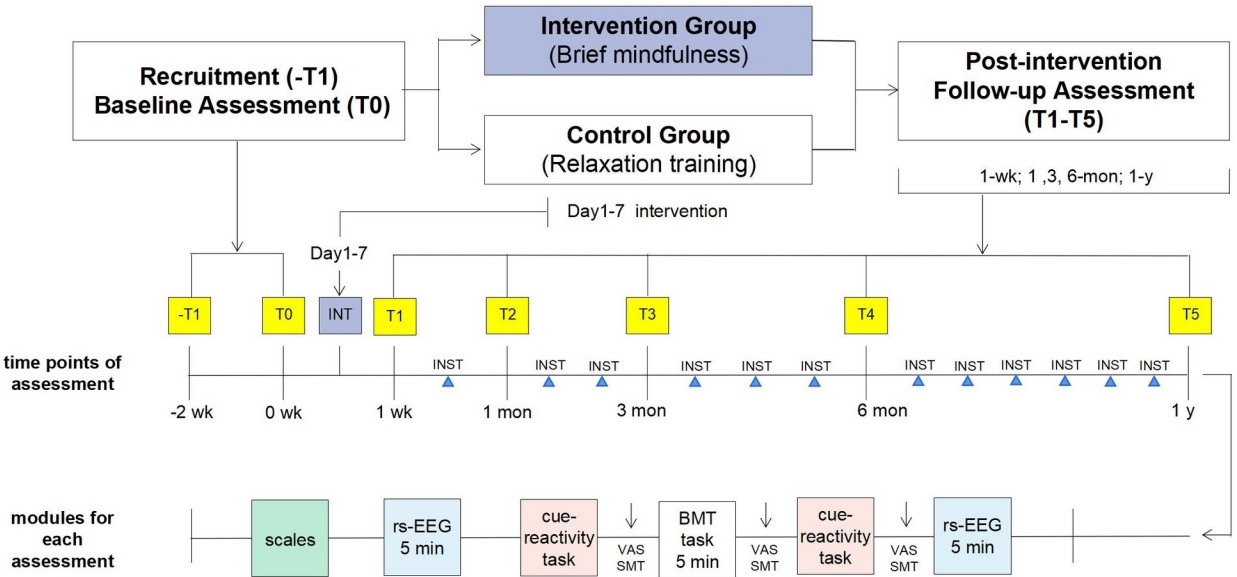

**Fig 3. Study design and procedure.** Note: The upper portion of the figure illustrates the overall study design. The middle portion depicts the timeline for assessing scales and EEG at baseline, post-intervention, and follow-up timepoints. Brief mindfulness and relaxation training will be administered to both the intervention and control groups after recruitment. The baseline assessment (-T1, T0) will be completed before the intervention. Audio guidance/learning material will be provided for daily practice in the brief mindfulness training and control group, a monthly intensive training session will be organized in follow-up period. The lower portion of the figure represents the modules of psychological and behavioral assessments during the baseline and five follow-up sessions. The cue-reactivity task will be implemented using a customized two-choice oddball paradigm. INT: intervention, INST: intensive training, rs-EEG: resting-state EEG; BMT: brief mindfulness training; VAS: Visual analogue scale; SMT: state-mindfulness.

anything: floating and conflicting mind, urging for cigarettes, transferred attention, sensations or tension in your body, or the sound surrounding, and again back to the feeling of air come into your nasal when breath.

**P = proceed;** Proceed with whatever you were doing before you came to a STOP or something that you want to do in the moment (e.g., proceed with thinking of the cigarette, or stop thinking about it and take an alternative behaviour, or even merely watching it disappears).

The STOP practice is brief and easy to remember and practice. As a most popular technique, we recommend STOP as a daily exercise based on basic mindfulness course. It will also be used in the later on study procedure of cue-reactivity task. The practice of STOP may cultivate a space between stimulus and response, which could help avoid the mostly spontaneous circle of trigger-behaviour-reward [65]. The adoption of 'STOP' exercise also takes into account that for beginners, the consistency setting of mindfulness practice methods in daily practice and cue-reactivity tasks may be beneficial in reducing differences caused by mastery and guidance of the technique.

## Cue-reactivity task

The experimental materials for this study are based on the image library proposed by Andrei Manoliu et al. [66]. The two-choice oddball paradigm will be adopted for it has the potential to offer dependable measurement indicators for assessing inhibitory behavior in individuals with addiction [67]. Moreover, the paradigm provides reaction time to understand the change of accuracy, which is absent in other tasks of tracking impulse inhibition (e.g., go or no-go task).

The reaction time and accuracy may indicate an attentional bias or objective resistance towards smoking cues.

The task would involve presenting a series of stimuli, with cigarette-related pictures serving as the target (oddball) stimuli among a series of non-smoking-related pictures (standard stimuli). Standard stimuli and deviant stimuli account for 80% and 20%, respectively. The experimental paradigm consists of 5 blocks: rest (5 minutes), smoking cue reactivity (approx. 15 minutes), STOP mindfulness audio (5 minutes) for managing somatic experiences related to smoking urges, another smoking cue reactivity block (approx. 15 minutes), and rest (5 minutes). The task involves 200 trials per block (40 deviant stimuli, 160 standard stimuli), with randomized presentation. Participants will have sufficient rest periods between blocks. The total time for completing the task is estimated to be approximately 15 minutes. Participants would be instructed to respond to both standard and deviant stimuli. The handedness of the button press was counterbalanced across participants. Event-Related Potentials (ERPs) activities that purely reflect behavioural control effects by subtraction techniques. By manipulating the image types of deviant and standard stimuli, it is possible to investigate the impact of cigarette related cues on the reaction inhibition process of smokers [67]. This paradigm will be created and presented using E-prime 3.0.

## Measures

The measures include smoking related indicators (craving, smoking behaviour, exhaled CO concentration, impulsivity), mindfulness level (state and trait mindfulness), affect (depression and anxiety), and records of daily practice for mindfulness. Relative spectrum power, microstate, phase synchronization indices and ERP components will be used to evaluate brain function (See Data analysis for details). The measurement tools and specific content are shown in Table 2. The outcome level and timepoints for assessment is annotated. Other individual information such as demographic data and social support levels will be measured. Online questionnaire links will be sent to the participants through WhatsApp at each time-point for them to fill in on their own devices.

## Data management plan

The data collected from all participants will be uniformly anonymized before statistical analysis. Data encryption or other secure storage strategy will also be taken. Firstly, we will strictly follow the standard operating procedures for different data collection methods in collection procedure. Collection and recording will be carried out by specialized personnel. All collected information will not be shared with any third party. Secondly, the source data will be stored on data platform and backed up multiple times (e.g., stored on non-networked hard drives or mobile hard drives specifically used by data analysts for data analysis); timely download and backup of raw data; backup software and regularly detect vulnerabilities, and prevent virus intrusion. As to the data quality, we will conduct double person data verification by checking the range of data values.

Data will be analysed after the completion of all stages of data collection without interim data analysis. The data generated for this study will be available from the corresponding author to researchers who meet the criteria for access to confidential data after the completion of the study.

Any information and data obtained about participants throughout the entire study will be treated with the utmost confidentiality. Unless participants give explicit permission, no information that could identify them will be shared with individuals outside the research team. Public reports on the study results will not disclose the personal identities of the participants.

Table 2. Evaluated variables.

| Variables | Timepoints | Measures | Outcome level |
|---|---|---|---|
| **Category 1: smoking related indices** | | | |
| Nicotine dependence | -T1, T0, T1-T5 | **Fagerstrőm test for nicotine dependence (FTND)** is a six-item revision of the Fagerstrőm Tolerance Questionnaire [69], scored as 0–3 (low), 4–6 (moderate), and 7–10 (high) with the summed score [70]. | primary |
| Craving | -T1, T0, T1-T5 | **1.Tobacco Craving Questionnaire-Short Form (TCQ-SF)** is short form of 12 optimized projects to assess smoking craving levels, including four factors of emotionality, expectancy, compulsivity, and purposefulness [71], with each factor scores from 3 to 21 points, was used in Chinese population [71]. | primary |
| | | **2.Questionnaire on Smoking Urges-Brief (QSU-Brief)** developed by Cox et al., [72] suitable for multiple measurements in laboratory and clinical use, including 10 items, can be completed in less than 2 minutes. Each item uses a subjective score of 1–7 points to evaluate the current level of craving for tobacco, the Chinese version will be adopted [74]. | |
| VAS for craving | -T1, T0, T1-T5, during CR task | **Visual analogue scale (VAS)** administered during cue-reactivity experiment to measure the state of craving, responses to each item were rated using a scale that ranged from 0 (not at all) to 10 (the strongest feeling possible). | primary |
| Number of cigarettes smoked/d | -T1, T0, T1-T5 | Self-report for number of cigarettes smoked per day. | secondary |
| Exhaled CO | T0, T1-T5 | Instantaneous reading device (Bedfont piCO Smokerlyzer) used to identify whether the duration of cigarette cessation meets the requirements. | secondary |
| **Category 2: mindfulness** | | | |
| Trait-mindfulness | T0, T1-T5 | **Five Facet Mindfulness Questionnaire (FFMQ)** consisting of 39 items and 5 factors, namely observation, description, acting with awareness, non-reactivity, and non-judging, the Chinese version will be used [75]. | primary |
| State-mindfulness | T0, T1-T5, during CR task | Based on the Toronto Mindfulness Scale (TMS), two items in Chinese version [76]were used to evaluate the dimensions of curiosity and decentralization. | primary |
| **Category 3: brain function** | | | |
| Brain function | T0, T1-T5 | Using a 64-channel EEG system to record in both resting and experimental conditions, with relative spectrum power, microstate, phase synchronization index and ERP components for evaluation. | primary |
| **Category 4: Socio-demographic and clinical information** | | | |
| Self-emotional assessment | T0, T1 | **Self-assessment manikin (SAM)** used to measure the emotional state during 7-day mindfulness intervention. | others |
| Affect | -T1, T0, T1-T5 | **Positive And Negative Affect Scale (PANAS)** compiled by Watson et al. [77], Chinese version [78] includes 10 corresponding adjectives that described either positive or negative emotion in both the both the negative and positive affect scales. Each project scores 1–5 points, with a total score of 10–50 points. to measure the emotional feelings of the past week. | secondary |
| VAS for affect | -T1, T0, T1-T5, during CR task | **Visual analogue scale (VAS)** for affect administered during cue-reactivity study to measure the state of affect, including valence and arousal. | secondary |
| Impulsiveness | -T1, T0, T1-T5 | **The Barratt Impulsiveness Scale Version 11 (BIS-11)** [79] was revised in 1995, consisting of a total of 30 items. The revised Chinese version [80] consists of 30 items, including three dimensions: motor impulsiveness, cognitive impulsiveness, and no planning impulsiveness, with a total score of 30–150, High scores represent hyperactivity, lack of concentration, and lack of planning, respectively. The higher the factor scores of each subscale, the stronger the impulsivity. | secondary |
| Record of daily practice for mindfulness | T0, T1 | Self-report of practice time; cognitive, emotional and behavioral states; course feedback and suggestions. | others |
| Adverse events | T0, T1-T5 | Proactive reporting without time constraints during research period. | others |
| Socio-demographic | -T1 | Age, gender, major, education year, Handedness | others |
| Handedness | -T1 | **Edingburgh Handedness Inventory (EHI)** is a measurement scale used to assess the dominance of a person's right or left hand in everyday activities, sometimes referred to as laterality [81], the Chinese version will be used [82]. | others |
| Depression | -T1 | **Patient Health Questionnaire (PHQ-9)** [55] self-administered tools for assessing depression for recent two weeks, with each project scoring 0–3 points and a total score of 0–27 points, with 6–9 points, 10–14 points, 15–21 points and 22–27 points indicating mild, moderate, severe and extreme severe depression. | others |

(*Continued*)

**Table 2.** (Continued)

| Variables | Timepoints | Measures | Outcome level |
|-----------|-----------|----------|---------------|
| **Anxiety** | -T1 | **Generalized Anxiety Disorder (GAD-7)** used to measure anxiety related problems over the past two weeks, with each project scoring 0–3 points and a total score of 0–21 points [57], with scores of 5, 10, and 15 corresponding to the cutoff values for "mild", "moderate", and "severe" anxiety levels [83]. | others |

*CR: cue-reactivity; CO: carbon monoxide.

Given the limited timeframe and the low known risk to participants in the current study, the formation of a Data Monitoring Committee is deemed unnecessary. However, the research team will remain vigilant in monitoring the situation and will reassess the need for establishing such a committee if circumstances change.

## Withdrawal from the program

Each participant can withdraw at any time during the study without giving any reason. Participants who voluntarily withdraw from the study will not be included in the analysis. The reason of withdraw will be recorded if the participants are willing to provide. It is important to emphasize that participant data may be used up to the point of withdrawal, unless individuals explicitly withdraw their consent for data usage.

## Safety considerations

In current study, all participants who encounter any health-related issues at any time are encouraged to actively contact with the researchers. The contact information will be provided during the recruitment or via group chats. Complete safeguard measures will be conducted even though there are very few procedures with possible safety hazards during the study. Firstly, we will continuously collect the health status information for each participant to assure their safety during the intervention training process. Any discomfort symptoms for participants, will be measured and handled immediately. We will directly contact with campus clinics if the participants report of severe physical or mental withdrawal symptoms.

The collection of EEG data will be carried out in a shielded room to isolate external electromagnetic and sound interference. There will be video devices inside the shielded room to synchronize the states of the participants to the researchers' terminal, to assure the safety when the participants in the room alone.

If, upon completion of the trial, it is determined that the stress level of any participant has reached the clinical threshold, active intervention will be offered by the psychological teachers at the on-campus psychological center. If required, participants will be referred to specialized hospitals for further intervention.

## Ethics and dissemination

This study was approved by Medical Ethics Committee, Kunming University of Science and Technology (approval number: Kmust-MEC-2023-004); see S2–S4 Files for registered and registering materials. Written informed consent will be obtained from each participant prior to participation in accordance with the Declaration of Helsinki and its later amendments or comparable ethical standards. The purpose, procedures and assessments, potential risks and benefits of the trial will be explained to all participants before recruitment. They will be informed

that their participation in this research is total voluntary after fully understanding the study. Moreover, participation can withdrawal at any time without reasons. Any changes or deviations from the protocol will need to be reported to the Medical Ethics Committee at the Kunming University of Science and Technology. This can be done by submitting a hardcopy application for amendment of an approved project form. Any results from this trial will be propagated via social media and be published in peer reviewed journals and conference proceedings.

## Data analysis

The current study aims to examine the efficacy of the intervention by using a two-arm parallel RCT design. For both the experimental and control groups, quantitative data, including scale measures and the EEG data, will be collected at baseline, post-intervention and four timepoints in follow-up procedure as mentioned above. All data will be analysed when collection is completed. The data analysis steps will also strictly follow the blind method principle. The R software (https://www.r-project.org/) and SPSS (IBM SPSS Statistics for Windows, V20.0), MATLAB (MathWorks in Natick, Massachusetts, USA) and EEGlab (https://sccn.ucsd.edu/eeglab) will be used for data analysis. Descriptive statistics will be applied for demographic, psychological and behavioural data in baseline, i.e., two sample t-test and $\chi^2$ test will be administered to continuous and categorical variables accordingly between BMT and CON group. For primary and secondary outcomes, repeated measures MANOVA will be performed to reveal the intervention effects while adjusting for individual factors and pre-intervention variables (e.g., trait mindfulness and smoking indices); within-group effects will be investigated through serial trend analysis from -T1 to T5. Data for all measures in different time points will be included in linear mixed-effects model analysis and growth curve model analysis to explore any factor associated with smoking cue-reactivity in different time-point for both groups. EEG data will be preprocessed using the Matlab software and the EEGLAB toolbox. The relative spectrum power for six EEG bands (delta: 1-4Hz, theta: 4-8Hz, alpha: 8-13Hz, low-beta: 13-20Hz, high-beta: 20-30Hz, and low-gamma: 30-48Hz) at both the single electrode level and regions of interest (ROI) level will be calculated. Microstate and phase synchronization analysis will be conducted. Functional changes involving subcortical brain regions will be investigated using source localization analysis.

For ERP data, ROIs comprising frontal, fronto-central, central and centro-parietal will be selected. The average wave amplitude of N2 (200-300ms) and P3 (350-550ms) components will be measured. ERPLAB toolbox will be used for all ERP component analysis. Additionally, the Fieldtrip toolkit will be utilized to calculate the phase lag index (PLI) [84] and phase locked value (PLV) [84]. A 2-second sliding window with a 50% overlap will be used to calculate the instantaneous phase difference for each sampling point. The frequency band of θ (4–8 Hz) α (8–13 Hz), low β (13–20 Hz) and high β (20–30 Hz) will be analysed. Both intra-band and cross-band coherence will be calculated. To avoid contamination from electromyographic signals, the γ frequency band will not be included in the analysis. The n*(n-1)/2 connection values will be calculated for each frequency band for all participants based on channel/data source. The individual matrices will be averaged and compared between groups to obtain the t-value matrix (see S4 File for more details).

Both repeated measurement analysis of variance and simple effect analysis will be conducted to EEG indices when considering the significance in professional field. Effect size will be measured using Cohen's d or partial eta-squared values [85]. missing data resulting from drop-out will be addressed with multiple imputation [87]. To obtain characteristic brain

functional change related to cravings, by correlation analysis of subjective cravings, behavior and EEG indicators.

The results of the aforementioned analysis will be reported following the CONSORT 2010 guidelines.

## The status and timeline

The participants of this study have been recruited and have completed some data collection. All subsequent studies will be completed in April 2024. We expect that all data analysis will be completed by the end of May 2024, all results will be announced by the end of June 2024, and the article will be completed by August 2024. The publication plan will consist of several sections, including the research plan, analysis of panel data using various statistical methods, analysis of psychology-related data, and conclusion.

## Discussion

The aim of the current study is set to explore the intervention effect of BMT on smoking cue-reactivity in college students with TUD. Specifically, we expect to explore how participants' responses to cues differ from their usual smoking behaviour by measuring their state mindfulness level, craving level, and EEG activity patterns before and after BMT. We ought to clarify the relationship between mindfulness improvement trajectory and subjective report of craving and to explore the intervention effect of BMT in reducing craving and smoking behaviour. Combined with EEG analysis, the RCT design will also be used to explore potential brain functional mechanisms. We will conduct EEG recordings during the cue-reactivity task to explore to what extent that EEG indicators may reflect this process and try to reveal the candidate dynamic models of brain function.

First and foremost, brief mindfulness usually defined in the duration of single and overall interventions, typically takes less than 20 minutes for a single training session, with a total intervention time of approximately one week, aimed at strengthening instantaneous attention and awareness [53]. However, compared with hospital patients, college student with TUD may have relatively weaker motivation to seek treatment, and the intervention effect of a single session may not reach the level of clinical patients. Therefore, on the basis of retaining the duration of a single intervention, we are planning to increase the duration of mindfulness exposure for participants through monthly intensive training and homework, to observe the long-term follow-up effect.

Additionally, the cue-reactivity paradigm is widely used as probe [88] for measuring addictive behavior, but there is significant heterogeneity in its application [88, 89]. In this study, we will employ a publicly available addiction image library and followed a standard two-choice oddball paradigm design. Response time and accuracy will be measured for both standard and deviant stimuli. Repeated measures ANOVA will be conducted to assess the reproducibility of the results and provide a valuable reference.

Lastly, previous studies on addiction-related EEG activity encompassed various indicators including time domain, frequency domain, brain regions, brain networks and so on. In this study, our main focus will be on comparing phase synchronization indices, power spectrum, microstate and ERP components that may be associated with cue-reactivity. For example, we aim to analyze how energy changes in specific frequency bands relate to neural activity regulation during smoking cravings. Moreover, by examining attention, cognitive processing, subjective reports, and behavioral cues related to cravings, we can evaluate the effectiveness of mindfulness interventions objectively.

## Strengths and limitations of this study

Although there are increasing researches on mindfulness among TUD population, most of them still limited to the adult population. By focusing the fragile also crucial period of brain maturation [90], our study is expected to yield several important findings.

Firstly, integrated analysis of behavioral and EEG records may provide new insights in investigating potential neurofunctional mechanisms of cue-reactivity in the young adults with TUD. The results can also enhance our understanding of pre-addiction brain changes, which will provide important supplements for different stages of addiction across all ages. The selection of subjective and objective indicators related to interventions are expected to provide clues for the development of targeted treatment strategies. Secondly, short-term mindfulness interventions align with the needs of the young people, the research findings of intervention setting could offer valuable guidance for practical applications. Moreover, The measurement of state mindfulness can track immediate effects, benefiting individuals in managing physical discomfort during cravings and mitigating stress and emotions triggered by daily life or cues.

However, there are still some limitations for the design. The primary consideration is as follows: 1) double blind method will not be used in this study, for the participants all from the same school, may inevitably discussing the experimental training they received. Moreover, the blind for experimental operators is also very challenging, as organization of the study need a lot of work in communication, would inevitably infiltrate by experimental related content. In order to mitigate the limitation (such as potential experimenter bias or participant expectations), the current study will involve a three-way evaluation that includes participants, psychological teachers, and experimental implementers, which will help partially address this issue. 2) the participants will be all in male, which limited the scope of analysis and generalization. As hormone indeed impacting brain functional status, the findings of brain mechanism research could restrict its generalizability across different genders. 3) biochemical indicators will not be measured with urinary cotinine, only through exhaled CO detection. As a result, false positives cannot be completely ruled out, and these measurements may not accurately reflect the actual level of nicotine metabolism. 4) no specialized behavioural measurement of intrinsic receptivity for mindfulness would be the Achilles heel for subjective research.

To address the challenges associated with double blinding and the absence of a data monitoring committee, the following procedures are necessary to uphold the integrity of blinding implementation: Impartial assessors will collect and evaluate data without prior knowledge of the study design to ensure unbiased evaluation. The data will be securely managed and analyzed using anonymized coding systems to maintain confidentiality. A clear separation will be maintained between the administrators of the intervention and the assessors of the outcomes to minimize bias. Robust standardized protocols will be implemented to promote consistency and minimize the risk of unblinding. Despite the challenges encountered, these measures are crucial for maintaining the integrity of the study. Furthermore, it is important to note that participants will not seek treatment. Although intensive training is about to be implemented, it is advisable to have lower expectations regarding training participation and effectiveness for this subgroup. However, it should be mentioned that the study will recruit volunteer who are not enrolled in psychological courses, which may indicate a certain level of enthusiasm among the participants. Therefore, different from smoking cessation plan, we will primarily focus on the intervention effect of BMT to cue-reactivity, and will emphasize brief mindfulness of STOP exercise training to facilitate the mastery of mindfulness for beginners. The audio guidance of STOP will also be used for online task of mindfulness in assessment to promote the homogeneity of research methods. The above factors are being of strength and weakness. Further studies with three-arm clinical design, adapting as full version of traditional training, or adding

effective component of mindfulness would need to be adopt if significant results from such a brief intervention provision promise.

To summarize, we prospect to explore whether participants' responses differed from their usual habitual behaviour of smoking after BMT and further assess the effects of mindfulness on response to urges. Possible biomarkers for this process and underlying mechanism in brain function will also be detected. The current study will present with reference value for studies in the similar population with higher smoking rates if the results of the current study proved to be effective.

## Supporting information

**S1 File. Copy of the study protocol approved by the ethics committee.**
(DOCX)

**S2 File. SPIRIT checklist.**
(DOCX)

**S3 File. All items from the world health organization trial registration data set.**
(DOCX)

**S4 File. EEG preprocessing and analyses.**
(DOCX)

## Acknowledgments

The authors are grateful to the Students Counseling and Mental Health Center for the contributions to the implementation of the study.

## Author Contributions

**Conceptualization:** Miaoling Luo, Quan Gan, Zhuangfei Chen.

**Data curation:** Linlin Cheng, Miaoling Luo.

**Formal analysis:** Quan Gan, Zhuangfei Chen.

**Funding acquisition:** Yu Fu, Zhuangfei Chen.

**Investigation:** Zhuangfei Chen.

**Methodology:** Linlin Cheng, Miaoling Luo, Quan Gan, Zhuangfei Chen.

**Project administration:** Zhuangfei Chen.

**Resources:** Zhuangfei Chen.

**Supervision:** Jie Ge, Yu Fu, Quan Gan, Zhuangfei Chen.

**Validation:** Quan Gan, Zhuangfei Chen.

**Writing – original draft:** Linlin Cheng, Miaoling Luo.

**Writing – review & editing:** Yu Fu, Quan Gan, Zhuangfei Chen.

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
