## [Decision Letter · Decision Letter 0]

23 Oct 2023

PONE-D-23-27631Effects of brief mindfulness training on smoking cue-reactivity in tobacco use disorder: study protocol for a randomized controlled trialPLOS ONE

Dear Dr. Chen,

Thank you for submitting your manuscript to PLOS ONE. After careful consideration, we feel that it has merit but does not fully meet PLOS ONE’s publication criteria as it currently stands. Therefore, we invite you to submit a revised version of the manuscript that addresses the points raised during the review process.

We look forward to receiving your revised manuscript.

Kind regards,

Lakshit Jain, MD

Academic Editor

PLOS ONE

 [This study was supported by the National Natural Science Foundation of China (NSFC) (Nos. 32060196, 82360271 and 82201597) and Yunnan Ten Thousand Talents Plan Young and Elite Talents Project (YNWR-QNBJ-2018-027, YNWR-QNBJ-2018-056), and Innovation team of Stress and disorder in nervous system in Yunnan.].  

a) If there are ethical or legal restrictions on sharing a de-identified data set, please explain them in detail (e.g., data contain potentially sensitive information, data are owned by a third-party organization, etc.) and who has imposed them (e.g., an ethics committee). Please also provide contact information for a data access committee, ethics committee, or other institutional body to which data requests may be sent. Please note that authors, including Corresponding Authors, are not permitted to be the sole point of contact for data requests.

b) If there are no restrictions, please provide the minimal anonymized data set necessary to replicate your study findings as either Supporting Information files or to a stable, public repository and provide us with the relevant URLs, DOIs, or accession numbers. For a list of acceptable repositories, please see http://journals.plos.org/plosone/s/data-availability#loc-recommended-repositories.

Reviewers' comments:

Reviewer's Responses to Questions

**Comments to the Author**

1. Does the manuscript provide a valid rationale for the proposed study, with clearly identified and justified research questions?

Reviewer #1: Yes

Reviewer #2: Yes

Reviewer #3: No

Reviewer #4: Yes

Reviewer #5: No

Reviewer #6: Yes

Reviewer #7: Partly

2. Is the protocol technically sound and planned in a manner that will lead to a meaningful outcome and allow testing the stated hypotheses?

Reviewer #1: Yes

Reviewer #2: Yes

Reviewer #3: No

Reviewer #4: Yes

Reviewer #5: Yes

Reviewer #6: Yes

Reviewer #7: Partly

3. Is the methodology feasible and described in sufficient detail to allow the work to be replicable?

Reviewer #1: Yes

Reviewer #2: Yes

Reviewer #3: No

Reviewer #4: Yes

Reviewer #5: Yes

Reviewer #6: Yes

Reviewer #7: No

4. Have the authors described where all data underlying the findings will be made available when the study is complete?

Reviewer #1: No

Reviewer #2: Yes

Reviewer #3: No

Reviewer #4: Yes

Reviewer #5: Yes

Reviewer #6: No

Reviewer #7: Yes

5. Is the manuscript presented in an intelligible fashion and written in standard English?

Reviewer #1: Yes

Reviewer #2: Yes

Reviewer #3: No

Reviewer #4: Yes

Reviewer #5: Yes

Reviewer #6: Yes

Reviewer #7: Yes

6. Review Comments to the Author

You may also provide optional suggestions and comments to authors that they might find helpful in planning their study.

Reviewer #1: The design and study aim are good but I would wait to submit until have atleast phase 1 data to prove something. Also there should be some reason provided for inclusion and exclusion criteria like why only rt handed etc.

Reviewer #2: Thank you for the opportunity to review the manuscript.

The study outlined a research plan aiming to investigate the effects of Brief Mindfulness Training (BMT) on smoking cue reactivity in college students with Tobacco Use Disorder (TUD). A two-arm parallel Randomized Controlled Trial (RCT) design, incorporating multiple data collection time points, various analytical software tools, and rigorous data analysis techniques. Ethical considerations and transparency are emphasized.

Strengths:

Robust Study Design: The choice of a two-arm parallel RCT is well-justified for evaluating the intervention's impact, and the inclusion of multiple data collection time points enhances the study's ability to capture temporal changes.

Comprehensive Data Analysis: The utilization of diverse software tools and detailed analysis plans demonstrates a methodologically rigorous approach. The inclusion of effect size measurements and the commitment to addressing missing data enhance the study's robustness.

Ethical Considerations: The commitment to reporting protocol deviations underscores the study's ethical transparency.

Future Research Directions: The inclusion of prospective avenues for future research provides valuable guidance for further exploration in the field.

Recommendations:

Literature Review: The study would benefit from a more comprehensive literature review to provide context for the research question and to highlight existing knowledge in the area of mindfulness interventions for smoking cessation and cue reactivity. This would help justify the research question and demonstrate how the study contributes to existing knowledge.

Discussion Section: While the discussion section outlines the study's strengths and limitations, it could be improved by providing a more in-depth interpretation of potential implications of the findings. Discussing how the study's results might advance the understanding of smoking cue reactivity and the role of mindfulness in college students with TUD would enhance the discussion's academic value and functionality.

Limitations: While the study acknowledges certain limitations, such as the lack of double-blind methodology and the male-only sample, it would be beneficial to discuss these limitations in more detail and explore their potential impact on the study's outcomes. Additionally, discussing strategies to mitigate these limitations could be valuable.

Reviewer #3: Given the regional focus, the introduction would benefit from more emphasis on the Healthy China Action, its purpose, aims, and the interventions it aims to accomplish, rather than on global tobacco concerns.

The technical descriptions of addiction and cue-induced responses require rewording and clarification. Many sentences appear to exhibit a form of circular logic. For instance, consider the following example: "It is assumed that addiction partially develops due to classical conditioning, where drug-related contextual cues become triggers that automatically prompt drug use." In this context, there is an organism affected by the continued use of a reinforcing stimulus that elicits a strengthened response, commonly referred to as habit strength. Furthermore, the chemical properties of nicotine function on brain pathways in a manner that reinforces negative behavior with continued use. It's worth noting that a cue can be regarded as a stimulus according to basic principles of behaviorism, and "trigger" is essentially another term for cue. Therefore, what is meant by an "automatic prompt" needs clarification. Classic principles can be employed to provide a more scientifically precise description of these concepts.

A thorough explanation of extinction learning is required, including how this process can occur in smokers. Furthermore, it remains unclear how cue-reactivity can be applied beyond the realm of classic stimulus-response terminology.

Mindfulness Training (MT) is not widely recognized as a commonly used treatment for Tobacco Use Disorder (TUD) – a citation is needed to support this statement, and any other statements where a claim is made on a fact.

Clear content organization is needed to address the organism that encounters a stimulus/cue and how the organism is positively

---

## [Author Response · Author response to Decision Letter 0]

8 Dec 2023

Dear Editor and Reviewers,

We would like to sincerely express our appreciation for the opportunity to address the valuable suggestions. The feedback provided has significantly helped to the improvement of our protocol optimization and the advancement of the research ideas. In response to these insightful suggestions, we have critically reviewed the manuscript, diligently incorporating the recommended revisions, with the aim of enhancing both the overall framework and the finer details. We are pleased to present a comprehensive revision that carefully considers the specific feedback provided. We kindly request the editor and reviewers to evaluate our revised manuscript accordingly. The manuscript has been thoroughly revised taking into account the valuable comments provided by the reviewers. We have attached the following files for your convenience: 'Response to Reviewers', 'Manuscript', and 'Revised Manuscript with Track Changes'. We kindly request your further review of these documents.

---

## [Decision Letter · Decision Letter 1]

9 Jan 2024

PONE-D-23-27631R1Effects of brief mindfulness training on smoking cue-reactivity in tobacco use disorder: study protocol for a randomized controlled trialPLOS ONE

Dear Dr. Chen,

Thank you for submitting your manuscript to PLOS ONE. After careful consideration, we feel that it has merit but does not fully meet PLOS ONE’s publication criteria as it currently stands. Therefore, we invite you to submit a revised version of the manuscript that addresses the points raised during the review process.

We look forward to receiving your revised manuscript.

Kind regards,

Lakshit Jain, MD

Academic Editor

PLOS ONE

Journal Requirements:

Additional Editor Comments:

Please address concerns raised by reviewer 8, and the article should be ready to accept once these additions are made.

Reviewers' comments:

Reviewer's Responses to Questions

**Comments to the Author**

1. Does the manuscript provide a valid rationale for the proposed study, with clearly identified and justified research questions?

Reviewer #5: Yes

Reviewer #8: Yes

Reviewer #9: Yes

Reviewer #10: Yes

2. Is the protocol technically sound and planned in a manner that will lead to a meaningful outcome and allow testing the stated hypotheses?

Reviewer #5: Yes

Reviewer #8: Yes

Reviewer #9: Yes

Reviewer #10: Yes

3. Is the methodology feasible and described in sufficient detail to allow the work to be replicable?

Reviewer #5: Yes

Reviewer #8: No

Reviewer #9: Yes

Reviewer #10: Yes

4. Have the authors described where all data underlying the findings will be made available when the study is complete?

Reviewer #5: Yes

Reviewer #8: No

Reviewer #9: Yes

Reviewer #10: Yes

5. Is the manuscript presented in an intelligible fashion and written in standard English?

Reviewer #5: Yes

Reviewer #8: No

Reviewer #9: Yes

Reviewer #10: Yes

6. Review Comments to the Author

You may also provide optional suggestions and comments to authors that they might find helpful in planning their study.

Reviewer #5: Thank you to the authors for addressing all of the comments. The manuscript is much stronger as a result.

Reviewer #8: This is an interesting study. However there are some fundamental information missing from the manuscript.

Some comments

Is the control group, usual care? As it seems to be another intervention? In which case should this be superiority/non-inferiority design?

For the sample size information, its unclear whether this sample size relates to one group, the experimental group or for both, in that is 45 the total?

Define your population of analyses,

In the data analysis section, can the authors state that results will be reported in accordance to CONSORT 2010 guidelines.

Inclusion of withdrawal of individuals in the study, be more explicit, unless the individuals withdraws consent of use of data, participant data an used at up to the point of withdrawal.

If there was no data monitoring committee appointed and the trial team reviewed the progress of the trial etc, how was blinding maintained?

Randomisation section, says matched for certain demographics, was this stratified randomisation?

Reviewer #9: Dr. Lakshit Jain, MD

PLOS ONE

Dear Dr. Lakshit Jain,

I have had the honor of reviewing the article titled "Effects of brief mindfulness training on smoking cue-reactivity in tobacco use disorder: study protocol for a randomized controlled trial” submitted to PLOS ONE for peer review.

The manuscript addresses a very important topic, psychological interventions to address tobacco use disorder, which is causing a significant burden on healthcare and one of the leading causes of global mortality rates. The authors stated that the current study proposed aims to explore the effect of brief mindfulness training (BMT) intervention on the smoking cue-reactivity in tobacco use disorder (TUD) among Chinese college students. The authors specifically targeted the smoking cue which is identified as an important trigger for relapse.

The authors of the manuscript hypothesized that brief mindfulness intervention can reduce smoking cue-reactivity in the population of TUD and can be reflected in subjective craving reports and changes in EEG functional indicators which they listed as their primary outcomes from this study adding secondary outcomes as daily smoking behaviors, affect and impulsivity, indicators reflecting correlation between mindfulness and smoking cue-reactivity. The authors further claimed that the study intends to explore the intervention effect of seven-day brief mindfulness practice on smoking cue-reactivity by a randomized controlled trial design, adding to the study the explanation of brain function mechanisms can also provide more targeted intervention strategies for smoking cessation.

To study the above association the authors employed a RCT design, assigned participants to a BMT group and a CON (control) group after gathering baseline data which included status related scales, EEG data for preintervention with the requirement for the participants not to smoke for at least three hours before collecting EEG data. At baseline the participants were required to report their last cigarette smoked and has to exhale CO concentration and only with those CO value below 10 to 7 PPM were asked to proceed to the EEG. Then the authors randomly assigned the participants to BMT and CON group, then they organized a seven-day mindfulness training guided by professional teachers specifically using “STOP” mindfulness exercises well the CON group carried out relaxation exercises. Postintervention data was collected when the training is completed and follow-up data in other four time points, specifically at one month, three-month, six-month and one-year along with the EEG data and skill data in the same time settings. The effects of mindfulness training were assessed using a series of linear mixed effects models. The authors used the image library proposed by Andre Manoliu where participants were asked to respond to both standard and deviant stimulate as part of the cue reactivity task.

The authors also disclosed the current status and timeline of the study protocol they proposed which stated that the participants of this study have been recruited and have completed some data collection, but subsequent experiments will be completed in April 2024 and all experimental data analysis will be completed by the end of May 2024 and results will be announced by end of June 2024 and article completion by the end of June 2024.

Some of the highlights of the study include the authors adopting a RCT design which removes the possibility of observer bias and to study the effectiveness of BMT in conjunction with obtaining EEG data which opens an avenue for further exploration of brain functioning mechanisms in TUD.

The authors appropriately identified multiple limitations including not using a double-blind RCT in the study along with identifying some demographic constraints including subjects being all male, not using urinary cotinine in conjunction with exhaled CO detection and no specialized behavioral measurement of intrinsic receptivity for mindfulness. Additional limitations include the participant population being confined to certain age group and a certain gender which may pose a challenge in application of the results to general population. Possible additional barrier to the results includes the application of the results based on the severity of TUD as the subjects were not screened if the current TUD is mild, moderate or severe. Though this is unique providing some references to existing data with similar studies and applications using EEG in substance use disorder would be helpful. Though may not be a limitation but needs attention is that the current study is under process and not complete.

Minor corrections include:

148: “Our previous trail” looks like the authors are referring to Trial and no trail.

Though the study does open the window in a specific direction and provide hypothetical novel approach of assessing brain functioning pre and post BMT and its effects on TUD, did not provide any conclusive data. Further research and data are needed upon conclusion of the study to build up on the findings and address the limitations comprehensively

Reviewer #10: All the comments seem to be addressed appropriately. Appears to be an interesting study and will be interesting to review results, when finished.

7. PLOS authors have the option to publish the peer review history of their article (what does this mean?). If published, this will include your full peer review and any attached files.

Reviewer #5: No

Reviewer #8: No

Reviewer #9: **Yes: **Surya Karlapati MD

Reviewer #10: No

While revising your submission, please upload your figure files to the Preflight Analysis and Conversion Engine (PACE) digital diagnostic tool, https://pacev2.apexcovantage.com/. PACE helps ensure that figures meet PLOS requirements. To use PACE, you must first register as a user. Registration is free. Then, login and navigate to the UPLOAD tab, where you will find detailed instructions on how to use the tool. If you encounter any issues or have any questions when using PACE, please email PLOS at <a href="mailto:figures@plos.org

---

## [Author Response · Author response to Decision Letter 1]

17 Jan 2024

Dear Editor and Reviewers,

Thank you for considering our manuscript for revision. We sincerely appreciate all the valuable feedback provided. In response to this feedback, we have carefully revised the manuscript to address the concerns raised and enhance its overall quality to meet the standards required for publication. We are grateful for the opportunity to further improve and strengthen our work. Once again, we thank you for your consideration and valuable input.

Journal Requirements:

Reply

We have carefully reviewed the reference list to ensure its accuracy and completeness. We have also added the DOI for each reference article, except for those that were not provided. Thank you for bringing this to our attention and your understanding.

Additional Editor Comments:

Please address concerns raised by reviewer 8, and the article should be ready to accept once these additions are made.

Reply

Thank you for your feedback and suggestion. We appreciate the specific concerns raised by reviewers and we have carefully incorporated the necessary additions to the article to meet the requirements. We appreciate your input and thank you for your time and guidance in improving the quality of our work.

Reviewer #5: Thank you to the authors for addressing all of the comments. The manuscript is much stronger as a result.

Reply

Thank you for your affirmation and valuable suggestions on enhancing the protocol.

Reviewer #8: This is an interesting study. However there are some fundamental information missing from the manuscript.

Some comments

Is the control group, usual care? As it seems to be another intervention? In which case should this be superiority/non-inferiority design?

Reply

Thank you for your thorough review and valuable suggestions. The study will employ a randomized controlled parallel trial design. In previous mindfulness research, relaxation training has been utilized as an active control (Tang et al., 2016). This training will involves watching documentaries showcasing natural scenery without any additional interventions. It has been suggested that video interventions with relaxing content can have a generic relaxation effect on autonomic regulation (Benz et al., 2022; Connolly, 2004). Hence, the inclusion of relaxation training as an active control in this study follows previous mindfulness research methodologies and enhances the study's superiority design. We have provided a description of the experimental design as follows: "Following previous mindfulness research methodologies, we will integrate relaxation training as an active control in this study to enhance its superiority design" (lines 226-228).

For the sample size information, its unclear whether this sample size relates to one group, the experimental group or for both, in that is 45 the total?

Reply

Thank you for pointing that out. Our previous statement may have lacked clarity. We have revised this section accordingly. Please refer to lines 279-282, where we clarified that "To ensure proper comparability, an additional 45 participants who match in demographic data and smoking severity will be assigned to the control group. Consequently, a total of 90 participants are required for the present study."

Define your population of analyses,

Reply

Thank you for your patience in reviewing our work. We have made the necessary supplements regarding the population of analyses as per your suggestion. For both the experimental and control groups, quantitative data, including scale measures and the EEG data, will be collected at baseline, post-intervention and four timepoints in follow-up procedure as mentioned above. All data will be analysed when collection is completed (line 554).

In the data analysis section, can the authors state that results will be reported in accordance to CONSORT 2010 guidelines.

Reply

Thank you for your suggestion. We have implemented the reporting method specified in the CONSORT 2010 guidelines in the data analysis section. Specifically, we have included the statement "The results of the aforementioned analysis will be reported following the CONSORT 2010 guidelines" (lines 596-597). 

Inclusion of withdrawal of individuals in the study, be more explicit, unless the individuals withdraws consent of use of data, participant data an used at up to the point of withdrawal.

Reply

Thank you for your valuable suggestions. We have made the necessary additions regarding the identification of subject withdrawal validity, as follows: "It is important to emphasize that participant data may be used up to the point of withdrawal, unless individuals explicitly withdraw their consent for data usage." (lines 515-516).

If there was no data monitoring committee appointed and the trial team reviewed the progress of the trial etc, how was blinding maintained?

Reply

Thank you for your reminder. We have made the following additions regarding the implementation of supervision. "To address the challenges associated with double blinding and the absence of a data monitoring committee, the following procedures are necessary to uphold the integrity of blinding implementation: Impartial assessors will collect and evaluate data without prior knowledge of the study design to ensure unbiased evaluation. The data will be securely managed and analyzed using anonymized coding systems to maintain confidentiality. A clear separation will be maintained between the administrators of the intervention and the assessors of the outcomes to minimize bias. Robust standardized protocols will be implemented to promote consistency and minimize the risk of unblinding. Despite the challenges encountered, these measures are crucial for maintaining the integrity of the study" (lines 681-690).

Randomisation section, says matched for certain demographics, was this stratified randomisation?

Reply

Yes, stratified randomization will be adopted to ensure a balanced distribution of important variables between the treatment groups, reducing the potential for confounding effects and enhancing the comparability of the groups. By matching participants in terms of demographic data and smoking severity, the study can help control for potential confounding factors that may influence the outcomes. 

Reviewer #9: Dr. Lakshit Jain, MD

Minor corrections include:

148: “Our previous trail” looks like the authors are referring to Trial and no trail.

Reply

Thank you for bringing this to our attention. We appreciate the correction, and we sincerely apologize for the spelling mistake in our article. The accurate phrase should indeed be "our previous trial" rather than "our previous trail". We will ensure that this correction is reflected in the revised manuscript. Once again, we apologize for any confusion that may have occurred (line 180). 

Though the study does open the window in a specific direction and provide hypothetical novel approach of assessing brain functioning pre and post BMT and its effects on TUD, did not provide any conclusive data. Further research and data are needed upon conclusion of the study to build up on the findings and address the limitations comprehensively

Reply

We appreciate your positive feedback. Currently, we are still in the protocol stage of our study. We value the input of your suggestion and hope to receive their suggestions to further enhance and refine our research plan through this submission.

Reviewer #10: All the comments seem to be addressed appropriately. Appears to be an interesting study and will be interesting to review results, when finished.

Reply

Thank you for your positive feedback. We appreciate your valuable suggestions for improving the protocol.

We would like to express our sincere appreciation once again for your valuable feedback. Your input has greatly improved the quality and rigor of our study, and we are deeply grateful for your support and valuable suggestions. Thank you for your dedicated efforts in reviewing our work and assisting us in enhancing our research.

Best regards,

Zhuangfei Chen

Email: chen.zhf@outlook.com

---

## [Decision Letter · Decision Letter 2]

16 Feb 2024

Effects of brief mindfulness training on smoking cue-reactivity in tobacco use disorder: study protocol for a randomized controlled trial

PONE-D-23-27631R2

Dear Dr. Chen,

We’re pleased to inform you that your manuscript has been judged scientifically suitable for publication and will be formally accepted for publication once it meets all outstanding technical requirements.

Kind regards,

Lakshit Jain, MD

Academic Editor

PLOS ONE

Additional Editor Comments (optional):

Reviewers' comments:

Reviewer's Responses to Questions

**Comments to the Author**

1. Does the manuscript provide a valid rationale for the proposed study, with clearly identified and justified research questions?

Reviewer #8: Yes

Reviewer #9: Yes

2. Is the protocol technically sound and planned in a manner that will lead to a meaningful outcome and allow testing the stated hypotheses?

Reviewer #8: No

Reviewer #9: Yes

3. Is the methodology feasible and described in sufficient detail to allow the work to be replicable?

Reviewer #8: Yes

Reviewer #9: Yes

4. Have the authors described where all data underlying the findings will be made available when the study is complete?

Reviewer #8: Yes

Reviewer #9: Yes

5. Is the manuscript presented in an intelligible fashion and written in standard English?

Reviewer #8: Yes

Reviewer #9: Yes

6. Review Comments to the Author

You may also provide optional suggestions and comments to authors that they might find helpful in planning their study.

Reviewer #8: All comments have been addressed.

Reviewer #9: Thank you for addressing the comments for the manuscript titled " Effects of brief mindfulness training on smoking cue-reactivity in tobacco use disorder: study protocol for a randomized controlled trial" submitted to PLOS ONE for peer review.

All the comments were appropriately addressed, and corresponding changes reflect the reviewers’ suggestions.

7. PLOS authors have the option to publish the peer review history of their article (what does this mean?). If published, this will include your full peer review and any attached files.

Reviewer #8: No

Reviewer #9: No
